# Prevalence and Risk Factors for Poor Sleep Quality in People Living with HIV: Preliminary Observations from an HIV Outpatient Clinic

**DOI:** 10.3390/v15081715

**Published:** 2023-08-10

**Authors:** Giuseppe Bruno, Massimo Giotta, Serena Perelli, Vito Spada, Maria Antonietta Purgatorio, Nicola Bartolomeo, Giovanni Battista Buccoliero

**Affiliations:** 1Infectious Diseases Unit, San Giuseppe Moscati Hospital, Azienda Sanitaria Locale Taranto, 74121 Taranto, Italy; serena.perelli@asl.taranto.it (S.P.); mariantoniettapurgatorio@yahoo.it (M.A.P.); giovannibattis.buccoliero@asl.taranto.it (G.B.B.); 2Interdisciplinary Department of Medicine, University of Bari Aldo Moro, 70121 Bari, Italy; massimo.giotta@uniba.it (M.G.); nicola.bartolomeo@uniba.it (N.B.); 3Infectious Diseases Unit, University of Bari, 70121 Bari, Italy; v.spada@studenti.uniba.it

**Keywords:** PLWH, sleep quality, PSQI, ART

## Abstract

Introduction: Sleep disorders have been reported in individuals living with HIV (PLWH), with a prevalence rate of over 50%. The main risk factors contributing to the development of sleep disturbances are not yet fully understood. We investigate the prevalence and risk factors associated with poor sleep quality in a population of PLWH who are receiving antiretroviral therapy (ART). Methods: The Pittsburgh Sleep Quality Index (PSQI) was used to evaluate sleep quality in PLWH attending our HIV Outpatient Clinic between October 2022 and April 2023. All subjects with a PSQI score > 5 were considered bad sleepers. A logistic regression analysis was carried out to assess risk factors associated with a PSQI score > 5. Results: A total of 132 PLWH (78% males) who received ART for at least one month were included in this observational study. The median age was 56 (IQR 47–61). Among all, 41 (31%) had a history of AIDS, and 95 (72%) were receiving an INSTI-based ART. The study population was divided into two groups: PSQI ≤ 5 (90; 68.2%) and PSQI > 5 (42; 31.8%). A lower BMI and the use of bictegravir in the current ART were associated with a PSQI score ≤ 5. In the multivariate analysis, the use of a bictegravir-based ART remained the only factor associated with better sleep quality (OR 0.17; *p* = 0.0222). No further associations between sleep disturbances and other epidemiological and clinical features were found. Conclusion: In this real-life scenario, poor sleep quality was observed in 31% of the cases, primarily among individuals with higher BMI. In addition, bictegravir users might seem to have a lower likelihood of experiencing poor sleep quality.

## 1. Introduction

Antiretroviral therapy (ART) effectively controls HIV replication and promotes immunological recovery, leading to an improved quality of life (QoL) for people living with HIV (PLWH) [1]. The safety profile of antiretroviral drugs in relation to the central nervous system (CNS) is crucial when selecting ART for both treatment-naïve and experienced individuals. Considering their high prevalence and strict correlation with QoL, the neuropsychiatric aspects play a predominant role in determining the appropriate antiretroviral regimen [2]. Sleep disorders are one of the commonly observed CNS disorders among PLWH, affecting over 50% of cases [3,4]. The international guidelines recommend investigating CNS disorders, including sleep disturbances, which are often underdiagnosed and underestimated [1]. These disturbances can impact the QoL and contribute to reduced adherence to ART [5]. The most commonly observed sleep disturbances in PLWH are insomnia, vivid dreams or nightmares, daytime fatigue or drowsiness and restless leg syndrome, and obstructive sleep apnea [6]. These disturbances not only affect daily physical and mental well-being but are also associated with metabolic imbalances and an increased risk of developing cardiovascular diseases [7]. The underlying mechanisms of sleep-wake cycle dysregulation in PLWH are still poorly studied and understood. It is hypothesized that HIV itself might trigger abnormal chronic immune activation, leading to the release of pro-inflammatory cytokines that could disrupt sleep homeostasis and contribute to these disturbances [8]. Indeed, receiving an HIV diagnosis can be an emotionally challenging experience, triggering mood disorders such as feelings of fear, anxiety, depression, and uncertainty about the future that altogether can contribute to the development of sleep disturbances or worsen sleep quality [9]. Sleep disturbances may manifest as difficulties falling asleep, staying asleep, or experiencing restful sleep, which, in turn, can lead to further emotional distress and exacerbate mood disorders. The combination of both the physical implications of living with HIV and the psychological effects of stigma and diagnosis disclosure can create a complex interplay that negatively influences sleep patterns and overall sleep quality [10].

Additionally, certain ART regimens in recent years have been associated with an increased risk of neuropsychiatric effects and disruptions in sleep quality [11,12]. The effects of antiretroviral therapy (ART) on the quality of sleep in PLWH can vary from individual to individual and depend on various factors, such as the type of antiretroviral drugs used, the duration of the therapy, the drug–drug interactions (DDIs), the overall health status, and the presence of other medical or psychological conditions.

Currently, several tools and questionnaires are used to assess sleep quality and designed to gather information about various aspects of a person’s sleep patterns, such as the Pittsburgh Sleep Quality Index (PSQI), Epworth Sleepiness Scale (ESS), Insomnia Severity Index (ISI), Berlin Questionnaire, STOP-Bang Questionnaire, Actigraphy and Polysomnography (PSG) [13,14,15].

These tools can be administered by healthcare professionals or used as self-assessment tools. They provide valuable information for diagnosing sleep disorders, monitoring treatment progress, and evaluating the effectiveness of interventions to improve sleep quality. Among all, the PSQI is a widely used and validated questionnaire designed to assess sleep quality in PLWH [16] as it can be easily administered during outpatient visits.

There is a lack of real-life studies that investigate the clinical factors contributing to poor sleep quality in PLWH receiving current ART regimens. Hence, our aim is to determine the prevalence and risk factors associated with impaired sleep quality, as measured by the PSQI, in PLWH.

## 2. Materials and Methods

### 2.1. Study Population and Data Collection

Our center takes care of over 450 PLWH and is one of the main Infectious Diseases Centers in Apulia, Italy. During the outpatient visit, we provide a review of the latest blood tests, measurements of weight and height, blood pressure monitoring, a physical examination as needed, and finally, the administration of the PSQI questionnaire to assess sleep quality. All consecutive individuals living with HIV (PLWH) who attended our HIV Outpatient Clinic between October 2022 and April 2023 and completed the PSQI questionnaire were included in this cross-sectional study.

The inclusion criteria were an age of 18 years or older and being on antiretroviral therapy (ART) for at least one month. Exclusion criteria consisted of PLWH who were not on ART for at least one month, subjects who refused to complete the PSQI, or those who did not provide informed consent.

For each participant, data were gathered and recorded in a dedicated and anonymized database, including the following information:Epidemiological data such as gender, age, nationality, and body mass index (BMI);Clinical data such as time from the HIV diagnosis, CDC staging, baseline CD4 and CD8 counts, nadir of CD4 cells, CD4/CD8 ratio, history of coinfections, history of virological failure to ART, viremia (HIV RNA) at baseline, current HIV RNA level, daily ART regimen, comorbidities, and concomitant medications.

A virological failure was defined in the presence of two consecutive HIV RNA ≥ 200 copies/mL [1]. A virological blip was defined as an isolated detectable HIV RNA level after suppression, followed by a return to HIV RNA suppression.

### 2.2. The Pittsburgh Sleep Quality Index (PSQI)

The Pittsburgh Sleep Quality Index (PSQI) is a self-report questionnaire that evaluates the presence of compromised sleep quality over a 1-month time interval. It is a relatively quick assessment, typically taking 5 to 10 min to complete. This straightforward questionnaire comprises 19 questions that address various aspects of sleep, categorized into 7 groups: sleep quality, sleep latency, sleep duration, habitual sleep efficiency, sleep disturbances, use of sleeping medication, and daytime dysfunction. Each component is rated on a scale from 0 to 3, and the scores are combined to obtain an overall global score ranging from 0 to 21. A higher PSQI score indicates poorer sleep quality and greater sleep disturbances. The validity of PSQI has been described by the authors as good, with a sensitivity of 89.6% and a specificity of 86.5% of patients versus control subjects. A score higher than 5 indicates poor sleep quality, as previously reported [16]. We used the Italian version of the PSQI [17].

### 2.3. Statistics

All data were anonymized and collected on an electronic database. Descriptive statistics were produced for demographic, clinical, and laboratory characteristics of cases. Mean and standard deviation (SD) were obtained for normally distributed variables, median and interquartile range (IQR) for non-normally distributed variables, and numbers and percentages for categorical variables. Univariate and multivariable logistic regression models were applied to evaluate the effect of the parameters (age, sex, BMI, comorbidities, current ART, time from the HIV diagnosis, time from the last ART, previous diagnosis of AIDS, HCV, HBV, and syphilis) on the probability of poor sleep quality. The results of the logistic models are expressed by the Odds Ratios (OR), their 95% Confidence Interval (95% CI), and the *p*-values of Wald’s tests. A *p*-value < 0.05 was considered statistically significant. Statistical analyses were performed by using the SAS/STAT^®^ Statistics version 9.4 (SAS Institute, Cary, NC, USA).

## 3. Results

A total of 132 individuals living with HIV (78% males) who were receiving ART and completed the PSQI questionnaire were included in this study. The median age of the participants was 56 years (with an interquartile range IQR of 47–61). Among them, 41 individuals (31%) had a history of AIDS, and 95 individuals (72%) were receiving an ART based on integrase strand transfer inhibitors (INSTIs). All subjects had a long history of HIV diagnosis (median 12 years). All subjects with viral hepatitis coinfections had HCV RNA and/or HBV DNA undetectable. Overall, the most common ART were the following combinations: the bictegravir-based single-tablet regimen (STR) in 33 subjects (25%) and a dual therapy based on lamivudine plus dolutegravir taken by 29 individuals (21.9%). Only one patient was receiving efavirenz at the time of the PSQI questionnaire. Furthermore, this patient obtained a PSQI score of 4. Thus, an efavirenz effect could not be fully evaluated in our population. Virological failures were reported as past events and were not ongoing at the time of the study.

The most common comorbidities were dyslipidemia (23.4%), cardiovascular diseases (22.7%), and metabolic disorders (15.9%). At least two comorbidities were observed in 28 subjects (21.2%).

The study population was divided into two groups: group A, consisting of 90 individuals (68.2%) with a PSQI score of ≤5, and group B, including 42 subjects (31.8%) with a PSQI score of >5.

The clinical characteristics of the two groups are summarized in Table 1. There were no significant differences between the two groups regarding demographic data, history of HIV infection, CD4 count, history of coinfections (viral hepatitis and/or syphilis), or the presence of comorbidities.

However, although not statistically significant, it was observed that individuals with a PSQI score > 5 had a lower percentage of undetectable HIV RNA (80.9% vs. 92.2%), a higher frequency of history of virological failure on ART (35.7% vs. 24.4%), a higher likelihood of neurological or psychiatric disorders (15.7% vs. 10.7%), cardiovascular diseases (26.3% vs. 23.8%), and dyslipidemia (31.5% vs. 22.6%). In total, eight individuals (6%) were taking central nervous system medications, including anxiolytics, antidepressants, antipsychotics, or anticonvulsants. Among these, three (3.3%) were part of group A, while 5 (11.9%) belonged to group B.

### Factors Associated with a PSQI Score > 5

To examine the risk factors associated with a PSQI score > 5, a logistic regression model was conducted (Table 2). In the univariate analysis, we observed that individuals with a greater BMI had a higher likelihood of having a PSQI score > 5. Additionally, bictegravir users showed a lower likelihood of experiencing poor sleep quality (OR 0.16; *p* = 0.0184).

After adjusting for sex and age in the multivariate analysis, a bictegravir-based ART remained the only factor associated with better sleep quality (OR 0.17; *p* = 0.0222). The average PSQI score in the bictegravir users was 4 (IQR 2–5). We noticed that among the bictegravir users bictegravir, the primary reason for an altered PSQI was waking up during the night to use the bathroom, occurring in 12 individuals (36.3%) at least once a week. Only three subjects receiving bictegravir reported experiencing nightmares at least once a week.

## 4. Discussion

Sleep disorders contribute to impairing the quality of life, worsening existing comorbidities or predisposing to their development, and are associated with reduced adherence to ART and poor clinical outcomes in PLWH [18,19,20]. In the past, certain antiretroviral drugs, such as efavirenz, have been associated with neuropsychiatric events, including impaired sleep quality [21,22,23]. Conversely, data regarding these effects of the current ART regimens on sleep quality are argued as drugs within the same class or of different classes can carry a diversified risk of neuropsychiatric events [24,25]. We assessed the prevalence of poor sleep quality using the PSQI questionnaire, which is simple, reproducible, and can be a useful tool during outpatient visits among PLWH. The PSQI has shown strong psychometric properties, to the extent that it can be considered a reliable and valid measure of sleep quality. It is capable of effectively differentiating between individuals with good and poor sleep patterns during screening, as well as identifying specific sleep disturbances experienced by individuals. The validity of PSQI has been described by Buysse et al. as good, with a sensitivity of 89.6% and a specificity of 86.5% of patients versus control subjects [16].

The risk factors associated with altered sleep quality in PLWH have been investigated in previous studies [3,11]. Ren et al. reported impaired sleep quality in 32.1% of the patients, a finding comparable to our study [11]. Furthermore, they found that the presence of anxiety and the use of an ART regimen based on efavirenz were associated with altered sleep quality, whereas family and social support played significant roles in improving sleep quality for PLWH. In a recent study, Mazzitelli et al. evaluated sleep quality using various tools, including the PSQI, in a cohort of 721 PLWH [26]. They assessed the presence of anxiety and depression using the Generalized Anxiety Disorder-7 and Patient Health Questionnaire-9. Interestingly, the study concluded that despite the higher prevalence of sleep disturbances in PLWH, these disorders were associated with the same determinants, such as cardiovascular risk factors and mood disorders, observed in the general population. They did not find a specific association between sleep disturbances and antiretroviral regimens or HIV-related parameters. Similarly, Petrakis et al. did not retrieve an association between the new ART regimens and sleep disturbances, unlike detectable viral load, low CD4 count, and limited physical activity, which were identified as significant risk factors for an increased rate of sleep disturbances [27].

Consistent with the findings from other studies [26,28], we did not observe significant differences in poor sleep quality in relation to HIV-related parameters. However, although not statistically significant, we noticed that individuals obtaining a PSQI score > 5 had a lower percentage of undetectable HIV RNA and a higher frequency of history of virological failure on ART. In this regard, two studies reported that higher sleep efficiency was correlated with lower HIV viral loads but not with CD4 count [29,30].

We found an association between poor sleep quality in subjects with a higher BMI. This finding has been described in the general population [31]. An elevated BMI and altered sleep quality can mutually influence each other in PLWH [32]. On the one hand, a high BMI can predispose individuals to sleep disorders, such as sleep apnea, which is characterized by breathing interruptions during sleep, leading to frequent awakenings and reduced sleep quality [33]. On the other hand, poor sleep quality can contribute to weight gain or even difficulties in losing weight [34]. Therefore, a multidisciplinary approach becomes crucial, involving the identification and treatment of any underlying medical conditions, addressing mental health concerns, reviewing medications that may impact sleep, encouraging a balanced diet and regular exercise, and tailoring optimal ART for each patient.

We reported that individuals who were undergoing treatment with a regimen that included bictegravir at the time of questionnaire administration exhibited a reduced likelihood of obtaining a PSQI score > 5 suggesting a potential protective effect of this treatment regimen against poor sleep quality. In fact, we observed that among the bictegravir users, the primary reason for an altered PSQI was waking up during the night to use the bathroom, occurring in 12 individuals (36.3%) at least once a week. Only three subjects reported experiencing nightmares at least once a week. Although this finding should be cautiously considered and needs further supporting evidence, some intriguing data from a clinical trial supported the beneficial effects of the use of B/F/TAF on patient-reported outcomes (PROs) [35]. In a recent large, longitudinal, real-world PRO study, a switch to B/F/TAF from previous regimens (mostly DTG- or EVG-based) in PLWH who were virally suppressed led to improvement 48 weeks after the switch in half of the 20 symptoms investigated [36].

Moreover, a study demonstrated that treatment-experienced PLWH eligible to switch their ART reported significant declines in the number and severity of DDIs if switched to BIC/FTC/TAF [37]. The number and burden of DDIs may influence sleep quality in PLWH. Therefore, it is of paramount importance to minimize the possibilities of pharmacological interactions that may disrupt the sleep-wake cycle, cause mood changes, and affect the state of alertness, potentially leading to reduced adherence to ART.

Currently, ART regimens based on INSTIs (Integrase Strand Transfer Inhibitors) are recommended as first-line treatments by international guidelines. However, although INSTIs have been associated with a risk of CNS side effects, the neurotoxicity profile varies among different drugs [2]. Specifically, a recent study reported that regimens containing DTG were associated with a higher rate of discontinuation due to reversible CNS-related events compared to regimens that did not include DTG [38]. A recent study evaluating sleep quality using PSQI in 119 PLWH did not find a correlation between sleep quality and treatment with bictegravir or dolutegravir compared to the other treatments [39]. Furthermore, a recent randomized, multicenter, open-label study demonstrated notable improvements in sleep quality, mood, and neuropsychiatric symptoms among 72 patients who were switched from dolutegravir/lamivudine/abacavir to darunavir/cobicistat/emtricitabine/tenofovir alafenamide. It is important to interpret these findings carefully and continue to conduct further research to better understand the nuances of the neurological effects associated with different ART regimens [40].

Our study has several limitations that should be acknowledged. Firstly, it was conducted at a single center with a small sample size, which may limit the generalizability of the findings to a broader population. Secondly, the evaluation of sleep quality solely relied on the PSQI questionnaire and did not involve the assessment of either QoL or PROs using other tools. Third, due to missing data, we could not assess the impact of the level of instruction, income, and level of employment on sleep quality in our study population. Fourth, as previously discussed, adherence to ART is a cornerstone in the management of HIV patients and may influence the occurrence of comorbidities and sleep disorders. In this study, we did not calculate the adherence to ART, but we believe it to be higher than 85%, as required for each patient during the outpatient visit. We hypothesize that the addition of sleep medications may be beneficial for patients with altered sleep quality. Unfortunately, in this study, we did not assess the effects on sleep quality resulting from the administration of anxiolytic or hypnotic drugs. It is crucial to analyze each patient’s sleep disorders, taking into consideration their medical history, current therapy, potential pharmacological interactions, and psychosocial conditions in their daily life, to individualize the management of sleep quality. Finally, the evaluation of sleep quality was performed during the outpatient visits, with a single administration of the PSQI questionnaire. Indeed, conducting multiple administrations of the questionnaire over an extended follow-up period, along with the use of other tools, could offer a more comprehensive understanding of sleep patterns and potential changes over time. Such an integrative approach allows us to gain a more in-depth understanding of how various factors, including the effectiveness of treatments, lifestyle changes, or psychosocial factors, may influence sleep quality over time. This enriched knowledge might lead to the development of targeted interventions and tailored strategies to improve sleep health and overall well-being for individuals living with HIV and beyond.

## 5. Conclusions

The quality of sleep in PLWH is often altered. We observed that subjects with a lower BMI and those taking bictegravir might seem to have a lower likelihood of impaired sleep quality assessed with the PSQI. These findings are worthy of future research to better optimize long-term ART for PLWH. Additionally, the effect on sleep quality may vary over time, and sometimes adjustments to the ART or supportive measures may be necessary to manage any sleep-related issues. To mitigate the negative effects on sleep quality, it is essential for PLWH to openly discuss any sleep problems they may experience during ART with their healthcare providers. Healthcare professionals can help identify and address the specific causes of sleep disturbances and implement appropriate interventions to improve rest and the overall well-being of PLWH. Therefore, we emphasize the concept that it is necessary to periodically assess sleep quality during outpatient HIV visits, recognize any disturbances, and then endeavor to address the underlying causes.

## Figures and Tables

**Table 1 viruses-15-01715-t001:** Clinical features of patients according to PSQI score.

		PSQI ≤ 5 N = 90		PSQI > 5 N = 42	
					*p*-Value
	N	% or IQR	N	% or IQR	
Age (years)	56	47–61	54	45–60	0.3434
Nationality					
Italian	87	97.75	38	92.68	0.3249
Foreigners	2	2.25	3	7.32	
Sex					
Male	73	81.11	30	71.46	
Female	17	18.89	12	28.57	0.2599
BMI, Kg/m^2^	24.45	22.60–26.40	26.4	23.9–29.80	0.0335
HBsAg positive	2	2.44	2	5.26	0.5903
Anti-HCV positive	17	20.73	6	15.79	0.623
History of Syphilis	13	16.25	7	18.42	0.7962
Time from HIV diagnosis (years)	12	5–24	12	6–23	0.6295
History of AIDS	24	30.38	17	45.95	0.1441
Time on ART (days)	6258	866–7124	5774	1770–7544	0.42
Time on current ART (days)	1254	479.5–1734	1372	690.5–2063	0.07
HIV RNA before the initiation of ART	72,355	9885–339,948	63,990	10,664–219,947	0.33
History of past virological failure to ART	22	24.4	15	35.7	0.179
HIV RNA undetectable at last control	83	92.2	34	80.9	0.05
Last CD4 cell count, cells/mm^3^	627	443–822	691	436–971	0.2417
Last CD8 cell count, cells/mm^3^	739	531–1039	804	626–1116	0.2717
Last CD4/CD8 ratio	0.83	0.54–1.29	0.77	0.48–1.44	0.8233
Current ART					
TAF/FTC + BIC	28	31.1	5	11.9	0.01
TAF/FTC + DRV/c	8	8.8	6	14.2	0.34
TAF/FTC + ELV/c	5	5.5	3	7.1	0.72
TAF/FTC + RAL	6	6.6	4	9.5	0.56
TAF/FTC + DTG	3	3.3	1	2.3	0.76
3TC + DTG	12	13.3	7	16.6	0.61
DTG + RPV	4	4.4	0	0	0.53
TAF/FTC + RPV	7	7.7	7	16.6	0.12
Other	17	18.8	9	21.4	0.73
Comorbidities					
Malignancies	4	4.4	2	4.7	0.934
Rheumatic diseases	1	1.19	1	2.63	0.5277
Neurological or psychiatric disorders	9	10.71	6	15.79	0.5522
Hematological diseases	1	1.19	1	2.63	0.5277
Cardiovascular diseases	20	23.81	10	26.32	0.8218
Dyslipidemia	19	22.62	12	31.58	0.3694
Renal diseases	1	1.19	1	2.63	0.5277
Respiratory diseases	2	2.38	0	0	1
Metabolic disorders	17	20.24	4	10.53	0.2997
At least two comorbidities	18	20	10	23.8	0.618

Abbreviations: BMI, body mass index; TAF, tenofovir alafenamide; FTC, emtricitabine; BIC, bictegravir; ELV/c, elvitegravir, cobicistat; DRV/c, darunavir; RAL, raltegravir; DTG, dolutegravir; 3TC, lamivudine; RPV, rilpivirine.

**Table 2 viruses-15-01715-t002:** Clinical factors associated with poor sleep quality (PSQI score > 5).

		Univariate			Multivariate	
	OR	IC95%	*p*-Value	OR	IC95%	*p*-Value
Nationality (Italian vs. Foreigners)	0.291	0.047–1.816	0.1862			
Sex (male vs. female)	0.582	0.248–1.365	0.2136	0.537	0.218–1.323	0.1763
BMI, Kg/m^2^	1.118	1.01–1.23	0.0321			
Age, years	0.981	0.95–1.012	0.222	0.977	0.945–1.01	0.1744
History of HBV (yes vs. no)	2.222	0.301–16.403	0.4337			
History of HCV (yes vs. no)	0.717	0.25–1.993	0.5235			
History of Syphilis (yes vs. no)	1.164	0.42–3.204	0.7691			
HIV RNA at baseline	1	0.993–1.008	0.9054			
CD4 cell count (last control)	1.001	1–1.002	0.1163			
CD8 cell count (last control)	1	0.999–1.001	0.7242			
CD4/CD8 (last control)	1.099	0.539–2.244	0.7945			
Time to first HIV-positive test	1.006	0.968–1.045	0.7782			
History of AIDS (yes vs. no)	1.948	0.871–4.357	0.1045			
Time from the initiation of ART	1	1–1.001	0.1389			
Type of ART						
INSTI users	0.495	0.224–1.091	0.0812			
PI users	1.697	0.683–4.217	0.2552			
NRTI users	1.055	0.306–3.642	0.9324			
NNRTI users	1.171	0.473–2.899	0.7327			
Use of Bictegravir (yes vs. no)	0.164	0.037–0.738	0.0184	0.172	0.038–0.777	0.0222
Comorbidities						
At least two comorbidities	1.962	0.818–4.703	0.1309			
Rheumatic diseases	2.243	0.137–36.84	0.5716			
Neurological or psychiatric disorders	1.563	0.514–4.755	0.4318			
Hematological diseases or Malignancies	2.243	0.137–36.84	0.5716			
Cardiovascular diseases	1.143	0.471–2.754	0.766			
Dyslipidemia	1.579	0.672–3.709	0.2944			
Renal diseases	2.243	0.137–36.84	0.5716			
Respiratory diseases	0.428	0.01–18.02	0.6567			
Metabolic diseases	0.464	0.145–1.486	0.1959			

Abbreviations: INSTI, integrase strand transfer inhibitor; PI, protease inhibitor; NRTI, nucleoside reverse transcriptase inhibitors; NNRTI, non-nucleoside reverse transcriptase inhibitors.

## Data Availability

The data that support the findings of this study are available from the corresponding author upon reasonable request.

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
