# Peer review of "Prevalence and Risk Factors for Poor Sleep Quality in People Living with HIV: Preliminary Observations from an HIV Outpatient Clinic"

_viruses, 2023, doi:10.3390/v15081715_

Round 1
Reviewer 1 Report
Thank you for giving me the opportunity of reading and reviewing your manuscript, I enjoyed reading the text and I have some general comments:
1. We have written a very similar short article. It would be interesting if you discussed some of the differences and cite the article (Vélez-Díaz-Pallarés M, Esteban-Cartelle B, Gramage-Caro T, Montero-Llorente B, Parro-Martín MA, Rodríguez-Sagrado MA, Ana María Álvarez-Díaz AM. Trastornos en la calidad del sueño asociados a los inhibidores de la integrasa en el tratamiento del VIH. Rev Esp Salud Pública. 2023; 97: 19 de junio e202306052). The main difference is that the only factor related to sleep disorders we found was the fact that the patients took medication to sleep or didn’t. We didn’t find differences between treatments, and we thought it was important to exclude efavirenz from the analysis. Could you address these issues?
2. Another issue you could consider is to include in your study is an analysis of which components of the PSQI are different between the group taking bictegravir and the rest. Maybe we find where the differences come from.
Thank you very mcuh
Author Response
REVIEWER 1
Thank you for giving me the opportunity of reading and reviewing your manuscript, I enjoyed reading the text and I have some general comments:
- We have written a very similar short article. It would be interesting if you discussed some of the differences and cite the article (Vélez-Díaz-Pallarés M, Esteban-Cartelle B, Gramage-Caro T, Montero-Llorente B, Parro-Martín MA, Rodríguez-Sagrado MA, Ana María Álvarez-Díaz AM. Trastornos en la calidad del sueño asociados a los inhibidores de la integrasa en el tratamiento del VIH. Rev Esp Salud Pública. 2023; 97: 19 de junio e202306052). The main difference is that the only factor related to sleep disorders we found was the fact that the patients took medication to sleep or didn’t. We didn’t find differences between treatments, and we thought it was important to exclude efavirenz from the analysis. Could you address these issues?
RESPONSE
- We sincerely thank the reviewer for their valuable suggestions. We have duly cited the study suggested, which is similar to our analysis and found no differences in antiretroviral therapies concerning sleep quality measured by the PSQI. In our study population, some patients were taking sleep medications or anxiolytics, as reported in the results. Specifically, eight individuals (6%) were taking central nervous system medications, including anxiolytics, antidepressants, antipsychotics, or anticonvulsants. Among these, three (3.3%) were part of group A, while 5 (11.9%) belonged to group B. It is important to note that the use of medications was a question included in the PSQI. However, we did not assess whether the intake of psychotropic drugs after the PSQI administration could have influenced the score. Only one patient was receiving efavirenz at the time of the PSQI questionnaire. Furthermore, this patient obtained a PSQI score of 4. Thus, an efavirenz effect could not be fully evaluated in our population."
REVIEWER 1
- Another issue you could consider is to include in your study is an analysis of which components of the PSQI are different between the group taking bictegravir and the rest. Maybe we find where the differences come from.
Thank you very mcuh
RESPONSE
Thank you for the question. Overall, Bictegravir was used in 33 subjects. The average PSQI score in this group was 4 (IQR 2-5). We noticed that among the bictegravir users bictegravir, the primary reason for an altered PSQI was waking up during the night to use the bathroom, occurring in 12 individuals (36.3%) at least once a week. Only three subject reported experiencing nightmares at least once a week. We added these data in the results.
REVIEWER 1
Thank you for giving me the opportunity of reading and reviewing your manuscript, I enjoyed reading the text and I have some general comments:
Thank you for reviewing the paper and for your time
- We have written a very similar short article. It would be interesting if you discussed some of the differences and cite the article (Vélez-Díaz-Pallarés M, Esteban-Cartelle B, Gramage-Caro T, Montero-Llorente B, Parro-Martín MA, Rodríguez-Sagrado MA, Ana María Álvarez-Díaz AM. Trastornos en la calidad del sueño asociados a los inhibidores de la integrasa en el tratamiento del VIH. Rev Esp Salud Pública. 2023; 97: 19 de junio e202306052). The main difference is that the only factor related to sleep disorders we found was the fact that the patients took medication to sleep or didn’t. We didn’t find differences between treatments, and we thought it was important to exclude efavirenz from the analysis. Could you address these issues?
RESPONSE
- We sincerely thank the reviewer for the valuable suggestions. We have duly cited the study suggested, which is similar to our analysis and found no differences in antiretroviral therapies concerning sleep quality measured by the PSQI. In our study population, some patients were taking sleep medications or anxiolytics, as reported in the results. Specifically, eight individuals (6%) were taking central nervous system medications, including anxiolytics, antidepressants, antipsychotics, or anticonvulsants. Among these, three (3.3%) were part of group A, while 5 (11.9%) belonged to group B. It is important to note that the use of medications was a question included in the PSQI. However, we did not assess whether the intake of psychotropic drugs after the PSQI administration could have influenced the score. Only one patient was receiving efavirenz at the time of the PSQI questionnaire. Furthermore, this patient obtained a PSQI score of 4. Thus, an efavirenz effect could not be fully evaluated in our population."
REVIEWER 1
- Another issue you could consider is to include in your study is an analysis of which components of the PSQI are different between the group taking bictegravir and the rest. Maybe we find where the differences come from.
Thank you very mcuh
RESPONSE
Thank you for the question. Overall, Bictegravir was used in 33 subjects. The average PSQI score in this group was 4 (IQR 2-5). We noticed that among the bictegravir users bictegravir, the primary reason for an altered PSQI was waking up during the night to use the bathroom, occurring in 12 individuals (36.3%) at least once a week. Only three subject reported experiencing nightmares at least once a week. We added these data in the results.

Reviewer 2 Report
Prevalence and risk factors for poor sleep quality in people living with HIV: preliminary observations from an HIV Outpatient Clinic
This article addresses risk factors associated with poor sleep quality in PLWH receiving ART and is based on preliminary observations from an HIV outpatient clinic. The study's primary finding is lower BMI, and adding bictegravir in ART improved sleep quality in PLWH. I found the study is meticulously designed and well executed. The present manuscript is well written. Following are the suggestions to improve the manuscript further,
1. As mentioned in the manuscript, the number of subjects is low, so there must be caution in concluding.
2. What was the effect of adherence to the ART regimen? Was there any known disruption? Was this higher frequency of virological failure on ART and a lower percentage of undetectable viral RNA leading to a PSQI score>5 due to disruption in ART?
3. Does adding sleep medication to ART could help improve sleep quality and the overall condition of PLWH?
Author Response
Dear Editor,
We are very grateful to the Reviewers for the useful suggestions and corrections, which will certainly contribute to improve the quality of our manuscript. All changes have been highlighted in the new version of the manuscript.
Our point-to-point response to the Referees’ observations are here reported:
